# Circadian Disruption and Mental Health: The Chronotherapeutic Potential of Microbiome-Based and Dietary Strategies

**DOI:** 10.3390/ijms24087579

**Published:** 2023-04-20

**Authors:** Pilar Codoñer-Franch, Marie Gombert, José Martínez-Raga, María Carmen Cenit

**Affiliations:** 1Department of Pediatrics, Obstetrics and Gynecology, University of Valencia, 46010 Valencia, Spain; 2Department of Pediatrics, University Hospital Doctor Peset, Foundation for the Promotion of Health and Bio-Medical Research in the Valencian Region (FISABIO), 46017 Valencia, Spain; 3Biosciences Division, Center for Health Sciences, SRI International, Menlo Park, CA 94025, USA; marie.gombert@sri.com; 4Department of Psychiatry and Clinical Psychology, Hospital Universitario Doctor Peset, University of Valencia, 46017 Valencia, Spain; martinezragaj@gmail.com; 5Microbial Ecology, Nutrition & Health Research Unit, Institute of Agrochemistry and Food Technology, National Research Council (IATA-CSIC), 46980 Valencia, Spain

**Keywords:** circadian rhythm, chronodisruption, chrononutrition, chronotype, time restricted feeding, diet, meal timing, microbiome and mental health

## Abstract

Mental illness is alarmingly on the rise, and circadian disruptions linked to a modern lifestyle may largely explain this trend. Impaired circadian rhythms are associated with mental disorders. The evening chronotype, which is linked to circadian misalignment, is a risk factor for severe psychiatric symptoms and psychiatric metabolic comorbidities. Resynchronization of circadian rhythms commonly improves psychiatric symptoms. Furthermore, evidence indicates that preventing circadian misalignment may help reduce the risk of psychiatric disorders and the impact of neuro–immuno–metabolic disturbances in psychiatry. The gut microbiota exhibits diurnal rhythmicity, as largely governed by meal timing, which regulates the host’s circadian rhythms. Temporal circadian regulation of feeding has emerged as a promising chronotherapeutic strategy to prevent and/or help with the treatment of mental illnesses, largely through the modulation of gut microbiota. Here, we provide an overview of the link between circadian disruption and mental illness. We summarize the connection between gut microbiota and circadian rhythms, supporting the idea that gut microbiota modulation may aid in preventing circadian misalignment and in the resynchronization of disrupted circadian rhythms. We describe diurnal microbiome rhythmicity and its related factors, highlighting the role of meal timing. Lastly, we emphasize the necessity and rationale for further research to develop effective and safe microbiome and dietary strategies based on chrononutrition to combat mental illness.

## 1. Introduction

Mental illness is currently one of the main leading causes of disability worldwide [1]. In fact, approximately 14% of the global burden of disease has been attributed to neuropsychiatric disorders. Recent estimations by the World Health Organization have drawn attention to the increase in mental illnesses not only in developed countries but also in developing countries [2]. The disabling nature of these conditions, as well as their strong association with many chronic physical diseases such as metabolic and immune-related comorbidities, highlights their importance. Limitations in the use of medications to treat mental health conditions are evident, with a considerable proportion of patients who do not respond appropriately to recommended treatments. In fact, the efficiency of medication varies greatly between patients with the same diagnosed mental conditions, which suggests the presence of diverse underlying mechanisms. In addition, there has also been little improvement in the primary prevention of mental illnesses. Therefore, new tailored alternatives are required for both the prevention and appropriate treatment of these disorders. A better understanding of the mechanisms implicated in their etiology and development is key to finding alternatives for both the prevention and appropriate care of these disorders. In this regard, a favorable lifestyle has been associated with an important reduction in the onset risk and a better prognosis for mental illnesses [3].

A common hypothesis of the recognized increased incidence of mental illnesses relies on modern prevailing lifestyle factors such as unhealthy dietary patterns, physical activity and sleeping habits, and the “stressogenic” environment in which we live. In this regard, the misalignment of the circadian rhythms (self-sustained rhythms with a period of approximately a day or 24 h) associated with modern lifestyles may make a significant contribution. Specifically, sleep–wake rhythms, the resulting chronotype, sleep quality, and circadian rhythm disturbances have been reported to significantly contribute to the vulnerability and persistence of symptoms of various mental disorders [4,5].

Across evolution, living beings have developed rhythmic behaviors and health reactions shaped by the temporal rhythm of our rotating planet. This temporal organization of life is so strategic for life that living species have developed their own organs to keep track of time. These organs are internal clocks, which in turn orchestrate behaviors and reactions to display circadian rhythms.

Circadian rhythms are regulated by the interaction of genetic [6] and environmental factors [7]. Among the environmental factors, lifestyle factors are major determinants of human circadian health [8]. They are precisely controlled by endogenous molecular clocks [9,10] that synchronize or align key internal behavioral and physiological processes at optimal times of the day based on cues from the environment or zeitgebers.

The suprachiasmatic nucleus (SCN) of the hypothalamus functions as a “central clock” or circadian pacemaker that interacts with the environmental light–dark cycle and coordinates “peripheral clocks” through various hormonal and neuronal signals sustaining the rhythmic gene expression of oscillating genes [9]. Specifically, approximately 5–10% of the genes in the peripheral tissues are regulated by circadian oscillations from the SCN, and they coordinate many crucial processes in the organism, such as immune function, homeostasis, and metabolism. Thus, body temperature, heart rate, and digestion, as well as attention capacity, mood, cognition, and behavior, present circadian rhythms, coupling biological functionality to the environment.

Although exposure to light is the primary environmental cue for circadian rhythms, feeding time and other environmental cues are basic for training and synchronizing peripheral clocks that, in turn, can influence the central clock. As described by Etienne Challet in “The circadian regulation of food intake”, the inner rhythms with respect to food confer two main advantages to individuals. On the one hand, there is the strategic temporal repartition of functions that should occur simultaneously, such as food intake and glycolysis, or activities that are meant to be performed at different moments, such as eating and sleeping. On the other hand, the internal clocks support an anticipatory capacity of the organism, for example, regarding the next time food will be available [11].

In the present review, we first describe the current scientific evidence of the relationship between disrupted circadian rhythms and mental illness. We also summarize the evidence that shows a bidirectional connection between gut microbiota and circadian rhythms, supporting the fact that the modulation of gut microbiota may aid in preventing circadian misalignment and in resynchronizing disrupted circadian rhythms. In addition, we refer to the diurnal dynamism of the gut microbiome and the related factors associated with diurnal oscillations of the gut microbiota composition and function, highlighting the role of food intake timing. Lastly, we emphasize the high potential impact of the daily control of feeding and fasting and gut microbiota on improving, via the resynchronization of circadian rhythms, the clinical management of neuropsychiatric conditions associated with circadian misalignment and/or for preventing mental illness in subjects at risk.

### 1.1. Clock Genes

Reactions presenting 24 h rhythms are encountered in all cell types, even in cells cultivated in vitro. These reactions result from the presence of oscillators called “peripheral clocks”. In fact, the circadian locomotor output cycles kaput (Clock), brain and muscle ARNT-like 1 (*Bmal1*), cryptochrome 1/2 (*Cry1/2*), period 1–3 (*Per1-3*), neuronal PAS domain protein 2 (*NPAS2*), timeless (Tim), and *Rev-erbα* are “clock genes” coding for proteins that together form a double loop of regulation over a period of approximately 24 h [10] (Figure 1). Under their control are numerous clock-controlled genes, within which there are transcription factors, receptors, transporters, and carriers, in addition to hormones, cytokines, and enzymes that play a key role in metabolism [12] and the immune system [13].

### 1.2. Central Clock

In addition to this inner time tracking system, external daylight is detected as a time cue by the retina. This signal is communicated through the optic nerve to the “central clock” located in the suprachiasmatic nucleus of the hypothalamus that is relayed to the pineal gland, which in turn inhibits the enzymatic process necessary for melatonin expression. When light exposure, especially blue wavelengths, decreases, this melatonin inhibitory process is lifted, and indoleamine is therefore produced and released into the fluids of the body [14]. In contact with cells, it can bind its specific membrane receptors from group G, enter the cell to bind cellular targets and nuclear receptors, or directly act as an antioxidant. Through these activities, melatonin regulates cellular metabolism in a circadian manner and acts as a synchronizer of the peripheral clocks [15].

Although human beings are diurnal animals, interindividual variability occurs with the preference to engage in activities earlier or later during the day. Accordingly, people have different chronotypes [16] that come from variability in the period of the internal clock. Early chronotypes (morningness) have a slightly shorter internal day, and late chronotypes (eveningness) have an internal day of longer than 24 h [17]. The chronotype has a genetic base, although it also depends on sex, evolves with age, and is shaped by exposure to external time cues [18].

### 1.3. Chronodisruption

External light, which regulates melatonin synthesis, is the strongest time giver perceived by the organism to synchronize with the external day. Overall, life habits such as nutritional intake, physical activity, sleep, social interactions, and mental and emotional activity form a network to influence circadian rhythms. These habits are particularly relevant to the timing, time amplitude, and quality and quantity of nutritional intake, which in turn affect and are affected by sleep quality, duration, and schedule [19].

Good synchrony of this system allows for the optimization of the organism’s functions to its needs. The opposite situation, called chronodisruption, is associated with numerous disorders [20]. Modern industrialized society has produced an important alteration in the relationship between host endogenous circadian rhythmicity and the environment [21]. With the advent of artificial light and industrialization, humans have undergone prolonged hours of light exposure and consequently have modified meal times, shifting the timing of food intake to later at night [22]. It has been revealed that this misalignment of circadian rhythms or chronodisruption [21] can have detrimental consequences for human physiological [23] and mental health [5]. Although the mechanisms by which disruption of circadian rhythmicity contributes to pathophysiological outcomes remain largely unknown, increasing evidence indicates that this misalignment of the circadian rhythms linked to a modern lifestyle may help explain the rising rates of many chronic diseases, including metabolic conditions such as obesity [24] and type 2 diabetes [25], certain types of cancer [26], and the high increase in the prevalence of psychiatric conditions [27,28].

Several clinical studies have suggested a strong association between mental health and circadian disruption [3,17]. However, the specific involvement of circadian health in the development and persistence of the most common psychiatric disorders, as well as the potential therapeutic benefits of circadian-based interventions, has been frequently overlooked [4,5,27]. Diverse pathophysiological factors may be involved, such as chronic circadian disruption that has been linked to a reduction in the complexity of neurons important for attention, cognitive flexibility, and executive function [28]. Circadian rhythmicity has recently started to garner great attention in neuroscience and psychiatry due to its impact on cognition, mood, and behavior. Moreover, the evening chronotype, commonly linked to circadian misalignment, has been described as a risk factor for more severe psychiatric symptoms and metabolic psychiatric comorbidities [29]. Impaired circadian rhythms, such as non-regular eating patterns and sleep disturbances, are commonly observed among individuals with various psychiatric disorders, including major depressive disorder (MDD), bipolar disorder (BD), anxiety, schizophrenia (SCZ), autism spectrum disorders (ASD), and attention deficit hyperactivity disorder (ADHD) [30]. Evidence indicates that targeting the resynchronization of circadian rhythms by making basic changes in lifestyle may improve the symptomatology of psychiatric conditions or even prevent the onset of mental health disorders in susceptible individuals. Additionally, data support the idea that the resynchronization of circadian rhythms may also reduce the risk of developing metabolic comorbidities in psychiatric patients [5].

### 1.4. Chrono-Nutrition and Nutritional Psychiatry

Chrono-nutrition is an emerging field in nutritional psychiatry based on the relationship between temporal eating patterns, circadian rhythms, and health [31]. In addition to dietary composition, the food intake window and the daily distribution of food intake are also critical factors for mental health [32,33,34]. In this regard, irregular and unhealthy meal timing has emerged as a key factor that can be disruptive to the circadian rhythm and mental health, and the benefits of clock-modulating diets are currently being increasingly recognized in psychiatry [35]. In fact, the evidence thus far supports that the circadian alignment of eating can prevent or improve internal circadian misalignment and consequently improve mental illness outcomes [33,34,35].

The gut–brain axis is a biological system that involves bidirectional interplay between the brain and the gut. The gut microbiota is a key regulator of this dialog by modulating various communication routes, including immune, endocrine, and neural pathways. Digestive physiology and intestinal barrier function are controlled by circadian clocks that also influence the expression of hormones and peptides, regulating nutritional intake through hunger and satiety sensations [11]. The gastrointestinal ecosystem has diurnal variation according to the state of food/fasting and the time of day, and these daily changes are reflected in the gut microbiota diversity and composition [36,37]. Healthy diurnal gut microbiota rhythmicity is largely modulated by food intake, and both the timing of food intake and the dietary composition shape the gut microbiota and are highly modifiable factors critical for circadian and mental health [35,36,37,38,39].

### 1.5. Circadian Disruption and Brain Development and Function

Circadian rhythms are important during the life course for regulating processes involved in the development of brain-related disorders [28]. In addition, there is consistent evidence highlighting that circadian rhythmicity disruption is one of the factors shared by most mental disorders across the lifespan, including neurodevelopmental-, cognitive-, mood-, and aging-related mental disorders [4].

The development of the circadian system begins in utero through the last prenatal period and continues slowly throughout the first years of life. Maternal circadian rhythms of feeding and core body temperature are important for the early development of fetal brain circadian systems [38]. Fetal tissues such as adrenal glands and the suprachiasmatic nucleus may be trained by the rhythms of maternal feeding schedules, glucocorticoids, and melatonin. In fact, training signals may cross the placental barrier to modulate or support the development of fetal circadian rhythms in the brain and peripheral tissues [28].

Remarkably, there is a robust link between circadian rhythm disruption during pregnancy, which is induced by maternal psychosocial and physiological stressors, and the development of neurodevelopmental disorders [38,39]. Maternal chronodisruption can impair cognitive and neurobehavioral outcomes in progeny. Gestational and early-life circadian disruptions have been associated with long-lasting negative consequences on offspring development and adult behavior in mice, including the emergence of social avoidance behaviors and hyperactivity in pups, even when they are cross-fostered by non-chrono-disrupted mothers, thus emphasizing the importance of circadian stability during pregnancy on offspring [40]. Furthermore, postnatal circadian disruption was also associated with reduced adult body mass, social avoidance, and hyperactivity [40]. Other studies have shown that even dim light at night has physiological and behavioral consequences on parents and offspring. For instance, chronic exposure of mice to dim light at night impairs their adaptive immune function [41] and increases depression-like behaviors in their offspring [42]. Pregnant rats exposed to simulated shift work have reduced weight gain during early pregnancy and reduced fat pad and liver weights, and they exhibit lower amplitude in the rhythms of their corticosterone, glucose, insulin, and leptin and impaired spatial memory in their offspring [43]. These memory deficits can be rescued by administering melatonin in drinking water to the mother, suggesting that maternal rhythms are important and can be modulated for correct early brain development of fetal circadian systems that can impact brain functioning [44]. These findings could be extrapolated to human populations, taking into account that circadian disruption during pregnancy easily results from modern lifestyle patterns. Maternal inflammation during pregnancy is posited to play a key role in the pathogenesis of neurodevelopmental disorders through its effects on fetal inflammatory and epigenetic pathways [45]. Increasing evidence supports the idea that environmental and lifestyle factors, including unhealthy diet, disturbed sleep, or microbial dysbiosis, contribute to systemic chronic inflammation in the mother with consequences for the developing nervous system [46], thus increasing the expression of neurodevelopmental disorders in childhood [45].

### 1.6. Circadian Disruption in Mental Health Disorders

Biological and clinical data have linked attention deficit hyperactivity disorder (ADHD), the most common neurodevelopmental disorder, with the disruption of circadian rhythms [47,48]. Reductions in sleep quality, delays in the circadian phase, and having an evening chronotype are consistently associated with ADHD in children and adults, and these alterations may be correlated with the severity of ADHD symptoms [47,49]. Specifically, ADHD is a condition with comorbid insomnia reported in >70% of children and adults, and a large proportion of patients with ADHD report emerging delays in sleep–wake rhythms, nocturnal rise in melatonin, and early morning rise in cortisol [47,50]. In addition, the high efficiency of chronotherapy approaches for the ADHD population strongly suggests that the circadian system may be an essential target for managing ADHD [49]. Furthermore, patients with ADHD have a high prevalence of obesity, and it has been proposed that the evening chronotype, a manifestation of circadian misalignment linked to ADHD and associated with unhealthy eating habits, may be an important mechanism linking ADHD to obesity [48]. Likewise, it has been reported that a majority of children with autism spectrum disorder (ASD) suffer comorbid sleep disturbances and have slower cortisol responses when compared with their typically developed sex-matched peers [51]. Moreover, polymorphisms in circadian-relevant genes have been associated with ASD [52].

There is a well-recognized bidirectional relationship between mood disorders and circadian rhythms. Mood disorders are often associated with disrupted circadian rhythms, such as sleep and melatonin and cortisol secretion, and the disruption of circadian rhythms can trigger or exacerbate affective symptoms in susceptible individuals [5]. Mood disorders are often characterized by sleep and circadian rhythm disruption or may be precipitated by an impaired light–dark cycle. Moreover, an impaired sleep pattern, which is commonly associated with disrupted circadian rhythms, is a diagnostic criterion for mood disorders (major depression, bipolar, posttraumatic stress disorders, and generalized anxiety). However, sleep disruption is not the only consequence or manifestation of circadian disruption that can influence mood and behavior. Studies conducted to elucidate the effects of circadian or sleep disruption on mood disorders using nocturnal animal species in which light at night does not directly alter sleep demonstrated that the impact of circadian disruption is observed, whereas sleep remains intact when animals are exposed to dim light at night [53].

Notably, rates of depression seem to correlate with the modernization of society, which implies a change in sleep routines. Circadian rhythm and sleep disorders can be precipitated by night, rotational, afternoon, or even early morning shift work schedules and are associated with irritability, depressed mood, and difficulties in maintaining personal relationships [5]. It has been reported that night shift workers suffer more health problems than their day shift worker counterparts [54]. Both shift work [55] and jet lag [56] or social jet lag [57] are associated with depression. However, while a meta-analysis of 11 studies concluded that night shift workers were 40% more likely to develop depression than daytime workers [58], other studies examining the association between MDD and shift work have reported contradictory results. A study with a representative sample of 3719 individuals from the general South Korean population reported that the prevalence of MDD among night workers was significantly increased relative to daytime workers [59]. In another population-based study conducted in Brazil that included 36,442 active workers, night shift work was significantly associated with MDD only in females [60], while there was no association between shift work and MDD in a further French study with a national random sample [61]. If all types of depression are combined, then a clear association emerges between shift work and depression. However, a study examining the amount of social jet lag in patients with MDD and healthy individuals reported no differences in social jet lag relative to healthy people and no association between social jet lag and the severity of depressive symptoms [62].

Interestingly, symptoms of depression often demonstrate diurnal variations, such that patients commonly exhibit symptoms in a morning-worse or evening-worse pattern, supporting a strong connection between MDD and circadian rhythms. Patients who typically express more severe symptoms in the morning more commonly show a more severe form of depression [63]. Furthermore, alterations in sleep/wake states, social rhythms, hormone rhythms, including reduced amplitude in melatonin and cortisol rhythms, and body temperature rhythms are observed in patients with MDD [64]. Clinical studies have also revealed that the severity of MDD is correlated with the degree of misalignment in circadian rhythms [65]. Adjunctive treatments to antidepressants, such as bright light therapy, wake therapy, and social rhythm therapy, have a positive effect on depression symptoms, suggesting the application of these approaches to circadian disturbances in the context of depression [66,67]. Thus, successful treatment of MDD with chronotherapies, as well as the abovementioned diurnal variation in depressive symptoms, supports the idea that circadian disruption may underlie the pathophysiology of mood disorders.

There is also experimental evidence about the association between disrupted circadian rhythms and depressive-like behaviors [68,69]. Two studies were conducted to assess the role of the SCN in regulating depressive-like behavior. Using the forced swimming test, an animal model of depression, it was concluded that bilateral destruction of the SCN had a protective effect during the induction of behavioral despair [70,71]. On the other hand, rats exposed to constant light exhibit increased depressive-like behavior accompanied by a loss of diurnal rhythms in physical activity and melatonin and corticosterone levels [69]. Interestingly, administering agomelatine, a melatonin MT1 and MT2 receptor agonist antidepressant, demonstrated the potential to alleviate depressive symptoms by improving biological rhythms in rats exposed to chronic constant light by modulating circadian rhythms. This approach would prevent the increase in depressive-like behaviors and restore diurnal corticosterone and melatonin rhythms [72]. However, it is difficult to extrapolate the findings of animal tests to human subjects as the circadian system of diurnal and nocturnal mammals differs.

Bipolar disorder (BD) is another mood disorder characterized by extreme fluctuations in a person’s mood, energy, and ability to function, alternating episodes of hypomania or mania and depression with periods of neutral mood [73]. Both human and animal studies have suggested an association between circadian disruption and BD [5]. Moreover, several polymorphisms in genes involved in the circadian molecular clock have been associated with BD, although with modest associations [74]. Additionally, jet lag by moving across many time zones induces bipolar episodes in vulnerable individuals; westbound travelers, who experience a phase delay in their circadian rhythms, have a higher rate of depressive episodes, whereas eastbound travelers, who experience a phase advance, are more predisposed to developing manic/hypomanic episodes [75]. Likewise, the disruption of circadian social rhythms, such as that of social jet lag, may also induce bipolar episodes that have been associated with mania but not with depression [76]. Chronotherapy with the aim to reset a dysregulated circadian rhythm could improve treatment strategies for BD. Early studies found that BD patients have a sleep–wake rhythm “fast-running” or abnormally short that likely leads to chronic circadian disruption, and they explained that treating with lithium, which slows the molecular circadian clock, is able to ameliorate the symptoms and stabilize circadian rhythmicity [77]. In addition, a randomized placebo-controlled clinical trial revealed that midday bright light therapy can resolve episodes of bipolar depression, thus again supporting that circadian health is a therapeutic target for BD [78].

The results of studies focusing on the association between circadian rhythm disruption and anxiety have been less consistent. The data have only provided modest evidence of this relationship [5]. Several studies have suggested that night shift work schedules and persistent jet lag may provoke anxiety. Studies in rodents revealed a relationship between the circadian system and anxiety-like disorders. In this sense, both the targeted disruption of canonical molecular clock components and the environmental disruption of circadian rhythms (e.g., via exposure to light at night) contribute to the development of anxiety-like behavior in animal experiments [79]. However, the effects of circadian disruption have been shown to be inconsistent across species [5], and the results suggest that they may depend on the developmental window during which circadian disruption occurs. For example, exposure to dim light at night during early development in mice increases adult anxiety-like responses, whereas exposing adult mice to light at night reduces anxiety-like responses [80].

Schizophrenia (SCZ) is a severe, chronic mental disorder characterized by disturbances in thought, perception, and behavior [73]. Given the extensive disruption of circadian rhythms in SCZ and the strong genetic component involved in this psychiatric disorder, it is not surprising that several polymorphisms located at circadian-related genes such as retinoic acid-related orphan receptor (*RORβ*), *Per2*, *Per3*, and neuronal Pas domain protein 2 (*NPAS2*) have shown putative associations that confer a sizeable risk for SCZ [81]. The severity of SCZ symptoms has been associated with the extent of sleep or circadian rhythm disruption, and circadian rhythm disruption is a common prodrome of SCZ [82]. Using postmortem brain tissues, it was revealed that subjects with SCZ lost rhythmicity in most genes within the prefrontal cortex that otherwise showed a rhythmic pattern of expression in healthy controls. By contrast, patients with SCZ displayed rhythmicity in a different set of genes that were not rhythmic in healthy controls, such as mitochondrial-related genes [83]. Furthermore, patients with SCZ have also been reported to show alterations in melatonin and cortisol, the two key endocrine transducers of the circadian clock [84]. In contrast to melatonin, the cortisol circadian rhythm appears to remain more or less intact during SCZ, although its concentrations are often elevated relative to those of healthy controls [85]. In addition, the altered regulation of the hypothalamic–pituitary–adrenal (HPA) axis in SCZ appears to be predictive of more severe psychotic and motor symptoms [86]. This hyperreactivity of the HPA axis is observed in both individuals with SCZ and individuals at high genetic risk for SCZ, which supports the model of stress vulnerability in SCZ [87]. Regarding the potential of circadian disruption as a part of the multifaceted picture in SCZ, the emergence and relapse of SCZ or schizoaffective disorder have been reported among individuals who have traveled across time zones [88]. However, there are no large controlled studies examining whether jet lag, social jet lag, simulated jet lag, or other forms of circadian disruption can precipitate or exacerbate psychotic symptoms.

Behavioral circadian rhythms decline in aged mice and humans [28]. Regarding neurodegenerative disorders, elderly subjects with dementia or Alzheimer’s disease (AD) are known to display cognitive decline and increased confusion and agitation in the evenings as the sun sets [89]. These individuals also show lower melatonin levels and a dysregulated HPA axis [89]. In fact, circadian dysfunction is a common symptom of AD [90] and a fundamental factor in aggravating the disease [91]. Although impaired circadian rhythms have long been considered a consequence of AD, recent evidence suggests a possible causative role for circadian dysfunction in its pathogenesis. The etiology and pathogenesis of disrupted circadian rhythms and AD share common factors, which opens the perspective of two mutually dependent processes, and epidemiological data show that impaired circadian rhythms in cognitively healthy adults are a significant risk factor for the future onset of AD [90]. Thus, all available data thus far indicate that addressing circadian disruption in AD may alleviate symptoms of the disease. However, to our knowledge, little is known about the application of chronotherapeutic strategies to target AD.

A growing body of evidence also points to significant alterations in the circadian system in Parkinson’s disease (PD). This is not unexpected, given the key role of dopamine in circadian regulation as well as the relevance of circadian influences in dopamine metabolism [28,92]. It is unknown whether circadian abnormalities might precede the development of PD. However, a cohort study showed that reduced circadian rhythmicity was associated with an increased risk of PD [93], indicating that it may represent an important prodromal feature for PD and suggesting that strategies to improve circadian function may influence the risk of developing PD and/or the disease prognosis.

## 2. Diet, Gut Microbiota and Mental Health: The Microbiota–Gut–Brain Axis

The gut microbiota, which is an intermediary nexus between the diet and the host physiology, has recently emerged as a key player controlling the recognized bidirectional communication between the gut and brain (the so-called gut–brain axis) [37,94,95]. In recent decades, numerous mechanisms have been described by which the gut microbiota, largely shaped by diet and by other factors such as birth mode, medication, and exercise, controls the gut–brain axis. This control is exerted through several mechanisms related to immune, metabolic, endocrine, and neural pathways that subsequently modulate brain development and function [96]. The gut microbiota plays numerous roles in physiological neurodevelopment, as well as in the development of the HPA axis that regulates stress responses, as shown in animal models. This crucial role suggests that modulating the microbiota–gut–brain axis may help protect against the onset and progression of mental disorders. Consequently, addressing the role of food habits in mental health has considerably changed the view of how neuroscientists understand the brain, cognition, and behavior. Consistent with this perspective, nutritional psychiatry is gaining attention based on scientific evidence. The notion of new dietary and microbiome-based strategies for improving mental health conditions [97] and the concept of psychobiotics as probiotics that confer mental health benefits have recently emerged, describing exogenous factors that influence the gut microbiota and have positive effects on mental health (Figure 2). Current scientific evidence highlights the impact of diet quality and other diet-related microbiota-targeted strategies, such as meal timing and psychobiotic supplements, on mental health (Table 1).

### 2.1. Bidirectional Link between the Gut Microbiota and Circadian System

It is well established that the relationship between the gut microbiota and host circadian rhythms is reciprocal. Circadian disruption induces gut dysbiosis, and gut dysbiosis itself can contribute, in turn, to chronodisruption [98]. Circadian clocks modulate the expression of hormones and peptides secreted from the gut. Both neural and endocrine afferent or efferent signaling appear to be involved in regulating nutritional intake through hunger and satiety control [11]. They also influence motility, luminal secretion, and mucus layer composition/thickness. All these elements result in a rhythmically changing niche for the microbiota according to bowel movements, the availability of nutrients, the gut barrier, and immune mechanisms [99,100]. Consequently, 24 h rhythms can be observed in the diversity and composition of the microbiota and in the produced metabolites that ultimately influence the host transcriptome [101].

The composition of the gut microbiota is entrained by host circadian rhythms and subjected to diurnal oscillation [102]. In this regard, animal studies have shown that circadian disruption affects microbiota communities [31,32]. In humans, this relationship has also been described. Circadian disruption leads to a shift toward potential proinflammatory taxa and a decreased abundance in microbiota-mediated pathways crucial for brain functioning, such as tryptophan biosynthesis for serotonin production [102]. Interestingly, some studies have reported that circadian disruption may alter the fecal microbial composition in a sex-dependent manner. In this regard, the absolute amount of fecal bacteria and the abundance of Bacteroidetes exhibited a circadian rhythmicity that is more pronounced in female mice [103]. Furthermore, the mouse microbiome is required for sex-specific diurnal rhythms of gene expression and metabolism and is, therefore, essential for sexual dimorphism [104]. A misalignment of the circadian system may cause alterations in the gut microbiota composition that can increase intestinal permeability and impair intestinal barrier function. As a result, there is an increase in the release of neuroactive compounds that act in brain regions responsible for mood [93]. Taking these facts into account, it can be suggested that circadian misalignment may adversely affect mental health through alterations in the gut microbiota composition and function.

Conversely, it is also clear that a diverse and healthy gut microbiota is essential for the optimal regulation of host circadian health [98]. Extensive research has described the action of the gut microbiota and its derived metabolites on the host circadian rhythm. Some gut microbes synthesize bioactive compounds that are highly involved in circadian health, such as serotonin, melatonin, or cortisol. In fact, a healthy circadian rhythmicity of these microbes should be regulated to ensure circadian health. Both germ-free and antibiotic-treated mice display alterations in their peripheral and gut clocks [101,104,105]. Additionally, germ-free mice showed a considerably altered expression of genes associated with rhythmic physiology, such as core circadian clock genes in the liver (Bmal1, Per1, Per2, and Cry1). Additionally, both germ-free and antibiotic-treated mice produce metabolites that do not oscillate diurnally [101], suggesting that the gut microbiota is crucial for metabolome rhythmicity. Moreover, diurnal gut microbiota rhythms drive the global programming of host circadian transcriptional, epigenetic, and metabolite oscillations. In relation to epigenetics, the presence of gut microbiota regulates diurnal oscillations and promotes rhythmicity in the expression of intestinal epithelial histone deacetylase 3 (HDAC3) [106]. By contrast, a lack of gut microbiota, as observed in germ-free mice, results in stably high HDAC3 levels. The oscillations in HDAC3 expression regulate histone acetylation and, consequently, are fundamental for metabolic gene expression in a diurnally oscillatory manner [106].

Gut microbiota-derived metabolites, including SCFAs (propionate, butyrate, and acetate) and biliary acids, modulate circadian rhythms. In vitro supplementation of hepatic organoids with butyrate or acetate leads to significant phase shifts in PER2 and mBMAL1 rhythms [107], and the oral administration of SCFAs to antibiotic-treated mice changed PER2 rhythms in peripheral tissue [108]. Other microbiota-related metabolites, such as unconjugated BAs, have also been shown to modulate circadian health [109].

Dietary quality changes may also markedly influence the gut microbiota’s diurnal rhythmicity. In fact, a high-fat diet can have detrimental effects on the microbiota, altering diurnal microbiota oscillations associated with a dysregulation of circadian clock rhythmicity [102,107]. By contrast, a healthy diet positively impacts the gut microbiota, and when coupled with a healthy functional gut microbiota, it encourages healthy host circadian rhythmicity [107].

The microbiota–gut–brain axis and circadian rhythms interact largely through the action of hormones secreted from endocrine glands controlled by the SCN. The two most prominent endocrine manifestations of circadian rhythms are the daily cycles of melatonin and glucocorticoids. These basic hormones for circadian rhythms can influence the gut microbiota, which in turn impacts their secretion by feedback mechanisms [110,111].

The pineal gland secretes melatonin, a key hormone in circadian rhythmicity. Melatonin is also synthesized in the gut by enterochromaffin cells and gut microbes, where it acts locally, as it is known as an antioxidant molecule [112]. Indeed, the concentration of melatonin in the gut is 400 times higher than that in the pineal gland. Light at night, acting by suppression of the melatonin rhythm, has the potential to alter physiology and behavior substantially [113]. Likewise, melatonin can train the circadian rhythms of some gut bacteria and modulate the chemotaxis of some microbes, such as *Escherichia coli*, together with specific components of the diet, such as polyamines [114,115]. Thus, melatonin may support a healthy gut microbiota that is involved in the maintenance of brain health. In this sense, melatonin can regulate neurodegeneration through the regulation of the circadian rhythm pathways in addition to the other mechanisms that can act on energy metabolism, epigenetics, or autophagy [14].

The circadian system also regulates the secretion of glucocorticoids, the end-products of the HPA axis, from the adrenal glands. Glucocorticoid concentrations tend to peak in the morning just before awakening and decrease throughout the day in diurnal animals and, inversely, in nocturnal mammals. The activity of the HPA axis oscillates under the control of the circadian clock, which is affected by changes in the gut microbiota [41]. Some bacteria can modulate this HPA axis, regulating the stress response [116]. Given the wide range of glucocorticoid effects on biological processes for survival stress response, it is not surprising that glucocorticoid concentrations are tightly regulated by negative feedback. However, dysregulated glucocorticoid secretion is associated with several health conditions, including major depression and anxiety disorders. The lack of a gut microbiome upregulated the glucocorticoid receptor pathway genes in the hippocampus, negatively influencing cognitive function and favoring depression [117]. Because light exposure is an important zeitgeber for the circadian rhythm of cortisol in humans, exposure to light at night could dysregulate the HPA axis, inducing hypercortisolemia, thus contributing to an increased risk of developing a depressive episode [118].

Loss of rhythmic host–microbiome interactions also disrupts the immune system [13] and consequently increases the risk of inflammatory and metabolic complications [50] commonly linked to mental health disorders [119]. Some studies have directly examined the circadian regulation of the immune response to gastrointestinal pathogens, which may impact mental health [120,121]. Scientific evidence indicates that immune dysregulation plays a key role in the pathogenesis of mental health conditions, at least in particular subgroups of patients such as those affected by multiple sclerosis or PD, and it highlights the potential implications of circadian reprogramming by dietary timing patterns to modulate immune-related diseases [122].

Gastrointestinal and metabolic comorbidities very often associated with mental health disorders [119,123] are commonly linked to circadian disruption [99] and gut microbiota alterations [124]. Therefore, exploring the connections between the microbiota–gut–brain axis and circadian rhythms can help to elucidate new potential therapeutic interventions (Figure 3).

### 2.2. Timing the Microbes: Circadian Rhythmicity of the Gut Microbiome

The gut microbiota is subjected to environmental changes such as nutrient availability and host-derived antimicrobial peptides. The gut microbiome is highly dynamic and exhibits daily cyclical fluctuations in composition and function [102,103]. In fact, a few years ago, Thaiss et al. reported the time-of-day-dependent composition of murine fecal microbiota [102]. The authors revealed that more than 15% of detected Operational Taxonomical Units (OTUs) undergo daily oscillations in their relative abundance, including those assigned to *Lactobacillus reuteri*, *Dehalobacterium* spp., and other species belonging to Clostridiales, Lactobacillales, and Bacteroidales. Other research groups also supported this observation. For instance, Zarrinpar et al. reported that 17% of OTUs showed diurnal cyclical oscillations [125], and Liang et al. noted an oscillation in the relative abundance of genera belonging to *Bacteroidetes*, *Firmicutes*, and *Proteobacteria* [103]. In addition to the fluctuation in relative taxonomic abundance, the bacterial genera differ according to circadian rhythms. The bacterial load in mice reaches a peak at 11 p.m. (active phase in nocturnal organisms), corresponding to a maximum in the Bacteroidetes population, and reaches a minimum at 7 a.m. (rest phase), corresponding to a maximum in the Firmicutes population [103,107,125]. Obesity is associated with a reduction in bacteria from the Bacteroidetes phylum and an increase in bacteria from the *Firmicutes phylum*. It is important to note that Bacteroidetes species rise during fasting and fall during feeding, thus reducing the Firmicutes/Bacteroidetes ratio associated with obesity.

Different microbial populations are subjected to changes based on food timing. Remarkably, the a-diversity (local species diversity) of the gut microbiota has been shown to increase with feeding and decrease with fasting [125]. Thus, an extensive number of gut bacterial genera and species, as well as the microbial community as a whole, exhibit oscillatory daily behavior in response to both the time of day and time of eating as well as the diet quality/composition [102,103,107,125]. Regarding meal timing, the diurnal rhythmicity of the gut microbiome is largely governed by host feeding/fasting fluctuations, and both regular time feeding and restriction of the feeding window can induce dramatic changes in the rhythmicity of the gut microbiota. Microbiome rhythmicity in night-fed mice shows a pattern opposite to that observed in day-fed mice, emphasizing the role of food intake in the diurnal oscillations of the gut microbiome. Lean mice fed ad libitum show strong microbial diurnal rhythmicity, which is dampened following a high-fat diet [125]. This observation shows that nutritional content is also an important factor for gut microbiota rhythmicity. However, when the mice were fed a time-restricted high-fat diet, they still maintained some gut microbiota diurnal rhythmicity and a lean phenotype [125], again highlighting the role of meal timing. Thus, time-restricted feeding (TRF) in nocturnal rodents, a form of intermittent fasting in which feeding is consolidated to the nocturnal phase, partially restores the cyclical fluctuations in the gut microbiota induced by a high-fat diet and protects against obesity and metabolic diseases. Another factor that has shown the ability to restore microbiome rhythmicity altered by a high-fat diet is the neurotransmitter melatonin. For example, melatonin specifically increases the magnitude of swarming in cultures of Enterobacter aerogenes but not in *Escherichia coli* or *Klebsiella pneumoniae* [112]. Furthermore, circadian disruption Per1/2/mice that were fed ad libitum lack gut microbial diurnal rhythmicity, which can be restored when these knockout rodents are assigned to a schedule of dark-phase time-restricted feeding [102]. Meal timing in humans has also been shown to affect the diurnal rhythms of the salivary microbial profile; eating the main meal late has been associated with increased salivary pro-inflammatory taxa [126].

Standard chow-fed mice that underwent a phase-shift representative of jet lag incurred a loss of diurnal rhythmicity in the gut microbiota [102]. Jet lag also exacerbates the effect of a high-fat diet in mice, specifically increasing weight gain and glucose intolerance. Jet lag in humans [102] also leads to a significantly altered microbiota composition characterized by a relative increase in Firmicutes. Conversely, metabolic health influences the rhythmicity of the gut microbiome. In this context, obesity or type 2 diabetes (T2D) displays less rhythmicity in the gut microbiome in ways that are distinct between the two disorders [127]. Sex is another additional factor that has been revealed to influence gut microbiome rhythmicity, likely via sex or growth hormones that are rhythmic and sexually dimorphic. In this regard, microbiome oscillations have been revealed to be more pronounced in female than in male mice [103].

Daily oscillations in the gut microbiota composition consequently imply functional oscillations [102]. Temporal changes in some pathways may help the gut microbiota anticipate changes in gastrointestinal functions, such as the availability of nutrients for growth. Moreover, the abundance of genes related to chemotaxis and flagellar assembly peaks during the resting phase, which may drive increased bacterial penetration into the mucus layer, facilitating the use of mucin as a nutrient [101]. These oscillations in microbial pathways in the gut lead to temporal changes in the levels of microbiota-derived and host metabolites. For instance, the butyrate and propionate levels are elevated in the cecum and feces in mice at the beginning of the light phase but remain lower throughout the rest of the day [107], and other metabolites, such as serotonin, ergothioneine, lysine, xylose, glucose, and isovalerate, also show significant oscillations across the day [101].

Currently, there is increasing interest in understanding how circadian rhythms could be used in the prevention and treatment of mental illness. Chronotherapy aims to restore the physiological circadian pattern, and chronopharmacology uses knowledge of biological rhythms to develop an optimal pharmacotherapeutic plan based on adjusting the time of drug administration to the times of drug metabolism, taking into account potential microbial drug transformations. Although it is well established that the gut microbiota and its secreted molecules follow a daily rhythm that interacts with the host and that the gut microbiota is well known to impact brain development and functioning, the potential disruption of microbiota rhythmicity linked to mental conditions has not been completely explored. The elucidation of the altered microbiome chronobiology linked to mental disorders may pave the way for highly innovative novel therapeutic approaches based on microbiome modulation. With this approach, further progress will be made in personalized medicine. Among the important actions that could promote or maintain an optimal circadian rhythm may be feeding control. Using specific experimental models directed to the different pathologies, the proposed hypothesis may be tested in the context of each mental condition. For instance, in relation to depression, the hypothesis could be analyzed using stress-induced depressive-like behavior as an animal model of depression and fecal microbiome transplantation from animals on intermittent fasting to recipient animals on a normal diet. This approach will allow the investigation of whether intermittent fasting ameliorating the depressive phenotype can at least be partially mediated by the gut microbiota.

### 2.3. Irregular and Mistimed Eating Patterns as a Disruptive Factor in Mental Health

Irregular meal timing is a disruptive factor in circadian rhythms and mental health [34,35]. It alters the timing of digestive processes involving the action of a variety of hormones that, in turn, regulate neurotransmitters influencing mood and energy at the neural level. Serotonin, a neurotransmitter that strongly impacts mood, is secreted primarily in the gastrointestinal tract. Daily oscillation in its production is controlled by a regular meal timing schedule, whereas irregular meal timing has been reported to break this pattern [35]. Irregular and mistimed eating patterns have been associated with depression, anxiety, and other psychiatric conditions [35], as well as with cardio-metabolic diseases, weight gain, and aging [59]. Shift workers, a group that commonly experiences irregular meal timing, have a 25 to 40% higher risk of depression and anxiety [54] and higher rates of other mental and metabolic health problems [128]. Eating on a regular schedule synchronized with circadian rhythms may reduce the risk and/or improve the prognosis of circadian-related disorders. Timing the food intake has now emerged as a novel strategy to minimize mood vulnerability in individuals experiencing circadian misalignment. Daytime eating can maintain internal circadian alignment and prevent glucose intolerance during simulated night work [129]. Furthermore, simulated night work with daytime and nighttime eating patterns increased depression-like mood levels by 26.2% and anxiety-like mood levels by 16.1% compared to the baseline, which did not occur with simulated night work in the daytime-only eating group. Notably, participants with a greater degree of circadian misalignment experienced more depression-like and anxiety-like levels during simulated night work, and the effect of the daytime meal timing intervention was associated with the degree of internal circadian misalignment [34]. Because circadian disruption alters microbiota communities, thus affecting mental health, an aspect to be explored is whether the benefit of daytime meal timing intervention may be achieved by maintaining gut eubiosis through circadian alignment.

### 2.4. Temporal Circadian Regulation of Feeding as a Novel Therapeutic Approach for Targeting Mental Illness

Temporal regulation of feeding and fasting has recently emerged as a potential innovative therapeutic strategy to prevent and/or treat mental disorders by modulating circadian rhythms and through other underlying mechanisms (Figure 4). Preclinical and clinical studies have indicated the potential benefits of intermittent fasting (IF) for epilepsy, AD, and multiple sclerosis on disease symptoms and progression of disease. Findings from animal studies have revealed some mechanisms by which IF may benefit Parkinson’s disease, ischemic stroke, AD, and mood and anxiety disorders [35]. Specifically, time-restricted eating exerts neuroprotective effects by upregulating the expression of neurotrophic factors and protein chaperones and by reducing the levels of proinflammatory cytokines. Fasting also appears to improve neurogenesis and synaptic plasticity, two essential mechanisms that can delay cognitive decay and neurodegenerative diseases.

There has been increasing interest in fasting in neuroscience and in the therapeutic potential of the ketogenic diet (KD) as an approach that mimics the effects of fasting for a variety of mental and metabolic disorders since the discovery early in the 20th century of the role of fasting-induced ketone bodies in epilepsy [130]. KD is considered the dietary intervention with the most proven therapeutic effect in nutritional psychiatry, especially for the treatment of refractory pediatric epilepsy. The mechanisms underlying the neuroprotective and anti-seizure effects of the KD have been linked with the gut microbiota in a study using two mouse models of treatment-refractory epilepsy [131]. The study showed that microbiota depletion via high-dose antibiotic treatment increased seizure susceptibility and incidence in response to the KD in mice, while these effects were reversed by recolonization with gut bacteria, suggesting that links between antibiotic use and seizure incidence in humans could be mediated by the gut microbiota. Consequently, the KD affects the gut microbiota, promoting selected microbial interactions that reduce bacterial gamma-glutamylation activity, decreasing peripheral gamma-glutamyl-amino acids, increasing the GABA/glutamate ratios in the hippocampus, and protecting against seizures. It was recently found that the gut microbiota during KD in epilepsy show taxonomic changes, such as the increase in the bacterial species *Akkermansaia muciniphila* and *Parabacteroides merdae*. It is of note that co-enrichment of these two bacteria (but not individually) in GF mice or pre-treated with antibiotics confers similar protection against epileptic seizures as the KD [131].

Ketones (acetoacetate, b-hydroxybutyrate, and acetone) are produced after at least 12 h of fasting as an alternative to glucose for the brain. They constitute an efficient source of energy and possibly enhance neuron bioenergetics and cognitive performance [132]. The increased b-hydroxybutyrate levels in hippocampal and cortical neurons induce the transcription of the brain-derived neurotrophic factor, a member of the neurotrophin family of proteins that is commonly decreased in depression and other psychiatric conditions. It plays a critical role in the development, maintenance, and plasticity of the central and peripheral nervous systems, and it has been demonstrated that subjects suffering from insomnia display significantly lower serum levels [133].

Furthermore, time-restricted feeding (TRF) has been shown to lead to a significant and durable improvement in sleep quality and efficiency [134]. TRF is a form of intermittent fasting in which all nutrient intake occurs within a few hours (less than 12 h) every day. It appears to be the most promising and interesting protocol in the field of neurosciences because it provides the benefits of fasting without altering nutrient quality or calories. Initially, the goal was to extend the time spent in a fasted state to prevent, postpone, and treat chronic diseases associated with aging [135]. However, an important question to address is whether or to what extent the (positive) effects of this type of intermittent fasting on several brain disorders, specifically on mental disorders, are triggered by the proposed fasting-induced metabolic, cellular, and circadian responses or by the reduced caloric intake often associated unintentionally with TRF. Daily fasting has been demonstrated to improve health and survival in mice regardless of diet composition and amount of energy [136]. In addition, the daily location of the fasting window seems to influence the response. In fact, evidence indicates thus far that the results of TRF largely depend on the time of day of the feeding window and are not only related to the fasting duration per se. In relation to metabolism, a feature highly linked to mental health, restricting food intake in the middle of the day was associated with reduced body weight, fasting glucose, hyperlipidemia, and inflammation [137], while the maintenance of the feeding period during the late afternoon or evening produced mostly null results or even worsened metabolic outcomes [138]. Thus, the circadian system may explain these dichotomous time-of-day effects. Consistent with chrononutrition principles, which essentially state that eating according to the circadian rhythm is crucial for circadian health and wellbeing, these results indicate that the beneficial effects of early time-restricted feeding, a form of intermittent fasting that involves eating early in the day to properly align food intake with circadian rhythms, are particularly relevant. In fact, restricting food intake to a specific appropriate daily interval seems to be crucial, even if the ingested diet is unbalanced, to synchronize some peripheral clocks and circadian rhythms [102].

Periodic fasting has been reported to modify brain neurochemistry and neuronal network activity to optimize brain function and peripheral energy metabolism [139]. This characteristic could be of particular interest to psychiatric patients with chronic circadian misalignment linked to immunometabolic alterations. Circadian disruptions associated with mental health disorders and metabolic disturbances are reciprocally linked, which suggests that targeting disrupted circadian rhythm and, consequently, metabolic disturbances may improve the psychiatric symptoms of patients with mental disorders. Furthermore, chronic inflammation associated with metabolic dysfunction, which is driven partially by dietary factors/habits, impacts mental illness development and aggravates the course of the disease. Regarding the timing of food intake, recent findings suggest that eating more frequently, reducing evening energy intake, and fasting for longer nightly intervals may lower systemic inflammation and subsequently reduce the risk of inflammation-related disorders. Recent evidence also suggests that circadian rhythms and meal timing may also play a role in the gut microbiota profile, with consequent potential effects on systemic inflammation and mental health outcomes. In this regard, TRF-induced improvements are accompanied by reduced levels of proinflammatory cytokines [140], which would explain the more favorable effects of TRF in patients with mental health conditions and comorbid inflammation.

Emerging research suggests that the key benefits of TRF on physical and mental health are largely mediated by the gut microbiota [125,141,142]. In relation to mental health, a recent study reported potent immunomodulatory effects of TRF that confer protection against multiple sclerosis and are, at least partially, mediated by the gut microbiome. Several authors specifically reported that TRF led to an increase in bacterial diversity, especially of the *Lactobacillaceae*, *Prevotellaceae*, and *Bacteroidaceae* families, and an enhancement of antioxidative microbial metabolic pathways. In addition, they also reported that TRF led to a reduction in IL-17-producing T cells and an increase in regulatory T cells [141].

## 3. Conclusions and Future Directions

Altered circadian rhythms are commonly reported among individuals with several psychiatric disorders, including those suffering from MDD, bipolar disorder, anxiety, ADHD, and SCZ. However, since most clinical data published thus far are correlational, the precise nature of the relationship between circadian rhythm disruption and psychopathology in the context of each mental condition is poorly understood. It is well known that circadian disruption increases disease severity, which in turn can worsen circadian rhythms. Rodent studies have demonstrated that induced circadian disruption can lead to affective changes, indicating that although circadian disruption may not be the single cause of mood disorders, it may elicit or exacerbate symptoms in individuals with a predisposition for mental health disorders. Conversely, the targeted resynchronization of circadian rhythms through different chronotherapeutic interventions in both humans and rodents has been demonstrated to improve symptoms of mood disorders and other psychiatric conditions. 

Although the precise mechanisms involved are still unknown, growing evidence supports the interaction and bidirectional communication between circadian rhythms and gut microbiota. Diurnal oscillations in the gut microbiome have emerged as a new feature of the microbiome with profound implications for circadian biology and human health. Currently, it is necessary to study the potentially altered microbiota compositional oscillations in the context of mental illnesses. Recent research has recognized that mistimed and irregular mealtimes associated with mental conditions may strongly impact diurnal microbiome rhythmicity. Because food intake is a major driver of circadian rhythmicity in microbiome composition and function, unhealthy meal habits commonly observed in subjects with mental disorders could possibly contribute via changes in the gut microbiota to psychiatric symptoms and metabolic disturbances that are frequently comorbid with mental conditions. Conversely, although the data are currently mostly derived from animal studies, they indicate that aligning the eating schedule with the circadian rhythms may prevent the development of mental health disorders in susceptible subjects or improve the clinical symptoms in psychiatric patients via the modulation of intestinal ecology. Furthermore, considering the links of the microbiome with the beneficial effects of intermittent fasting as well as the impact of regular circadian fasting on daily microbiome rhythms, it is plausible that at least some of the beneficial effects of circadian fasting or other types of time-restricted feeding interventions on mental health may be due to an enhancement of the rhythmic properties of the microbiome. Thus, future studies are required to establish whether changes in meal timing can help individuals suffering from mental health disorders and to identify biomarkers, including those related to the microbiome, that could predict the benefit of time-restricted eating interventions in specific patient populations.

Subjects suffering from mental health disorders often find it difficult to adhere to dietary interventions. Thus, there is an urgent need to decipher the mechanisms by which dietary intervention exerts beneficial effects in patients with mental disorders, with a view toward designing safer and easy-to-implement targeted intervention strategies. The discovery of time-restricted feeding-induced gut microbiota signatures driving improvements in cognition and/or behavior could help to design future innovative tailored microbiota-based intervention strategies for targeting mental illnesses (i.e., prebiotics, multi-strain consortia, and postbiotics) that may also be further explored in combination with other lifestyle interventions, such as physical activity and cognitive or psychological interventions (Figure 5). This holistic approach may influence on the endocrine, immunological, microbial, and neural states responsible for improving mental health. Likewise, the metabolic and functional aspects of microbiota and the mechanisms by which they exert beneficial effects need to be elucidated to develop personalized and integrative nutrition/medicine strategies that could help to reduce the prevalence and burden of mental health disorders.

## Figures and Tables

**Figure 1 ijms-24-07579-f001:**
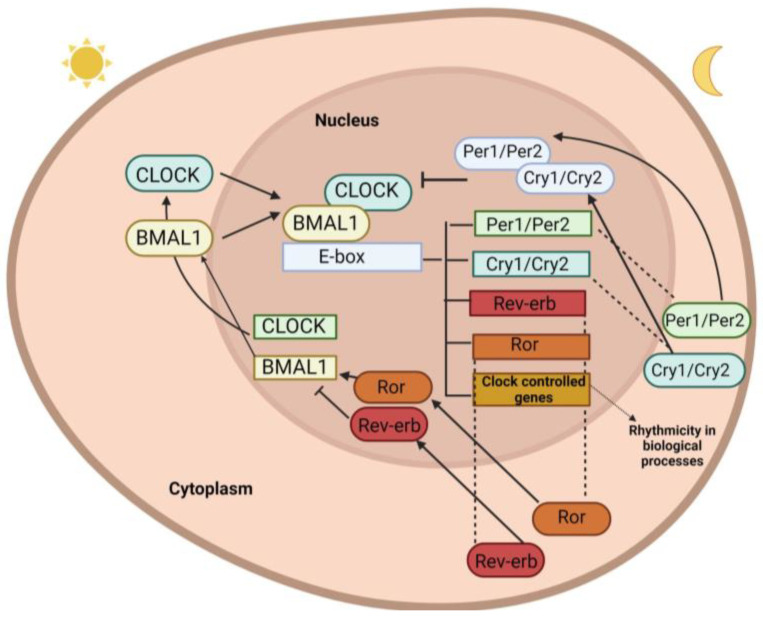
The molecular circadian clock mechanism. On a molecular level, the circadian clock is regulated by transcription factors that, upon excessive rising, inhibit their own expression through transcriptional–translational feedback loops. Specifically, the transcription activators BMAL1 and CLOCK heterodimerize in the cytoplasm and translocate to the nucleus to bind to the E-box promoter region of the genes *Per1/2* (period) and *Cry1/2* (cryptochrome), driving their transcription and translation. The translated proteins PER1/2 and CRY1/2 then heterodimerize and initiate inhibitory feedback of CLOCK and BMAL1, thus consequently inhibiting their own expression due to CLOCK and BMAL1 repression. Because ubiquitin degrades the PER-CRY dimer, the cycle begins again through the disinhibition of *BMAL1:CLOCK*. Additionally, the transcription factor REV-ERB suppresses *BMAL1*, and ROR (retinoic acid-related orphan receptor) activates *BMAL1*. This action generates a negative feedback loop in which gene expression, hormonal secretion, and protein levels oscillate on an ~24 h basis.

**Figure 2 ijms-24-07579-f002:**
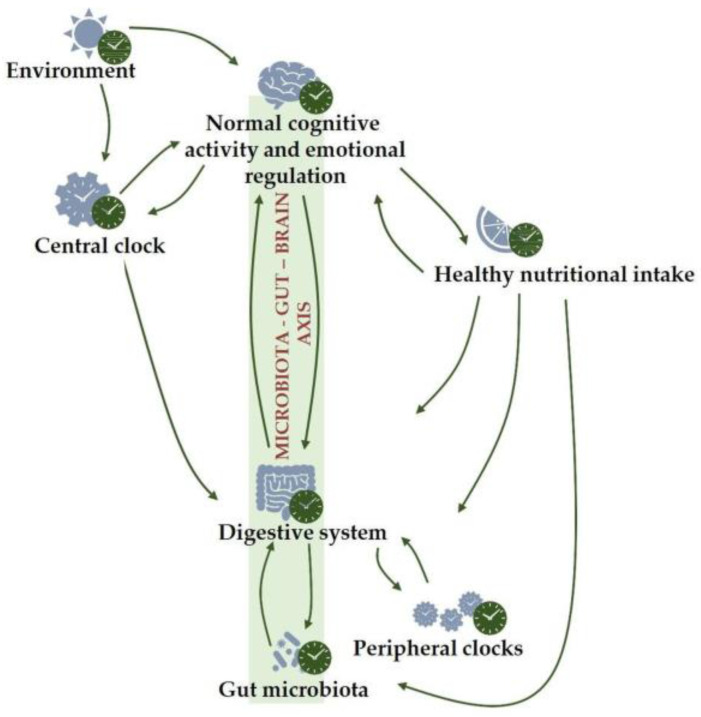
Depiction of circadian rhythms, the gut microbiota, and their interaction and effect on gut and mental health in healthy individuals. Circadian rhythms strongly influence cognition, mood and behavior, and brain function, which in turn, strongly impact circadian health. Food intake that is temporally coordinated by the brain over the circadian (~24 h) cycle impact on brain functioning largely by shaping gut microbiota and intestinal homeostasis.

**Figure 3 ijms-24-07579-f003:**
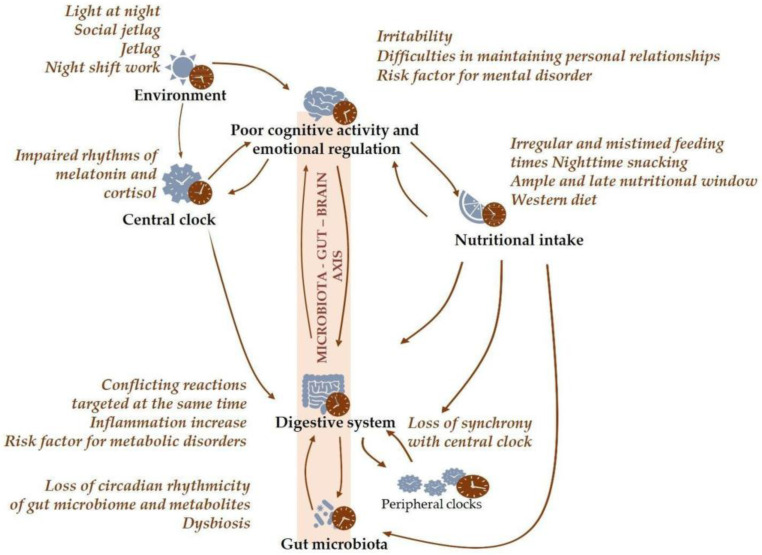
The tight bidirectional link between gut microbiota and circadian system and the impact of disturbances to circadian rhythms or the gut microbiota on host physiology and mental health. Although genetics influence circadian rhythms, lifestyle is a major determinant of circadian health, and factors linked to modern lifestyles can lead to chronodisruption, which often has detrimental consequences on gut microbiota and host physiology, including brain function.

**Figure 4 ijms-24-07579-f004:**
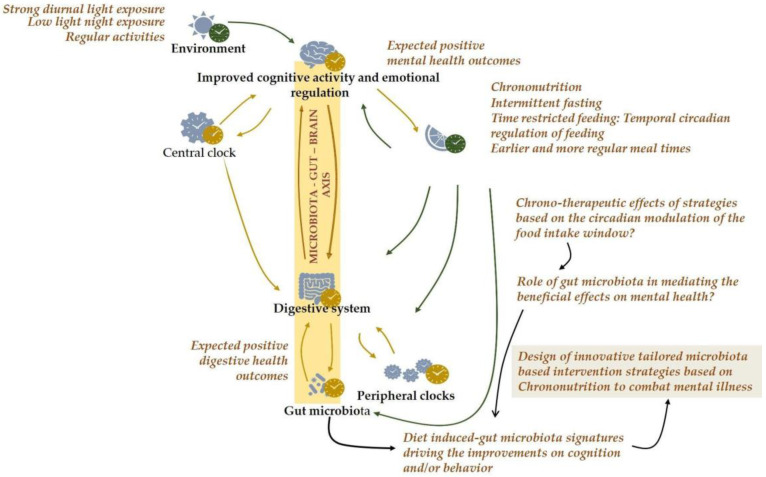
The advised high potential of chrononutrition-based strategies among the approaches for targeting the circadian system in the prevention and care of mental disorders. Irregular and mistimed eating pattern has been pointed out as a disruptive factor for mental health. The discovery of Chrononutrition-based strategies induced-gut microbiota signatures driving improvements on cognition and/or behavior could help to the design of innovative tailored microbiota-based intervention strategies for targeting mental illness. These microbiota-based strategies could also be further explored in combination with other lifestyle interventions, which may have potentially significant implications for personalized nutrition/medicine in terms of reducing the prevalence and burden of mental health disorders.

**Figure 5 ijms-24-07579-f005:**
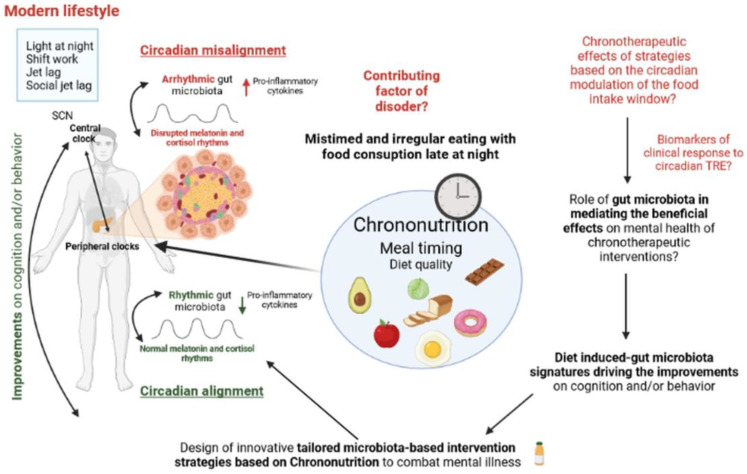
Potential prevention or restoration of circadian misalignment through microbiome-based and dietary strategies founded on chrononutrition. Food intake is a major driver of circadian rhythmicity in microbiome composition and function, and unhealthy meal habits, such as irregular and mistimed eating patterns commonly observed in subjects with mental disorders, could possibly contribute via changes in gut microbiota to psychiatric symptoms and metabolic disturbances. Aligning the eating schedule with the circadian rhythms may prevent the development of mental health disorders in susceptible subjects or improve the clinical symptoms in psychiatric patients via the modulation of intestinal ecology. The elucidation of the altered microbiome chronobiology linked to mental disorders and the role of chrononutrition in the restoration of the healthy microbiome chronobiology may pave the way for highly innovative novel preventive and therapeutic approaches. TRE: Time-restricted eating.

**Table 1 ijms-24-07579-t001:** Summary of dietary modifications of gut microbiota potentially affecting mental disorders.

Diet Quality	Outcome
High-fat diet	- Alteration of gut microbiota composition and function. - Lower bacterial diversity. - Reduced bacterial richness. - Disturbed diurnal gut microbiota oscillations. - Induction of Bmal1 and Clock gene expression during the dark phase. - Reduction of short-chain fatty acids.
High-fiber diets or short-chain fatty acids-containing diets	- Induction of peripheral clock adjustment. - Affected neurotransmitter metabolism with implications for enteric and central nervous system function. - Decreased gut dysbiosis. - Exert anti-inflammatory action.
Mediterranean diet	- Increased diversity of microbiota - Increase in short-chain fatty acid-producing commensals. - Increase in species known to confer mucosal protection and anti-inflammatory effects.
Ketogenic diet	- Restored GABA/glutamate balance. - Protected mitochondrial function. - Reduced overall alpha diversity, while increasing the relative abundance of Akkermansia muciniphila, a producer of short fatty acids.
Meal timing (Chrononutrition)	Outcome
Time-restricted feeding	- Maintenance of gut microbiota diurnal rhythmicity. - Restored gut microbiota composition altered on high-fat diet.
Intermittent fasting	- Increased gut microbiota diversity. - Enrichment of the Bacteroidaceae, Lactobacillaceae, and Prevotellaceae families. - Alters metabolic pathways of the gut microbiome: increased relative abundance of the synthesis and degradation of ketone bodies and glutathione metabolism pathways.
Psychobiotic supplements	Outcome
Probiotics/Prebiotics/Synbiotics	- Anti-inflammatory action suppressing immune and sympathetic reactions. - Ameliorates alterations in the circulating levels of pro- and anti-inflammatory cytokines that directly affect brain function. - Support of the gut barrier integrity.
Omega-3 fatty acids	- Proposed to restore a eubiotic state by increasing bifidobacteria and decreasing enterobacteria. - Enhanced formation of neurites and synapses. - Modulation of the serotoninergic and dopaminergic neuroendocrine transmission involving brain activity and sleep behaviors.
Polyphenols	- Positive modulation of circadian clocks. - Able to exert anti-inflammatory activity at the neuronal level.

## Data Availability

Not applicable.

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
