# Peer review of "Circadian Disruption and Mental Health: The Chronotherapeutic Potential of Microbiome-Based and Dietary Strategies"

_ijms, 2023, doi:10.3390/ijms24087579_

Round 1
Reviewer 1 Report
This is a review manuscript written with the aim to describe the link between circadian disruption and mental illness and to summarize the connection between gut microbiota and circadian rhythms, supporting the idea that gut microbiota modulation may aid in preventing circadian misalignment and in the resynchronization of disrupted circadian rhythms.
The manuscript is well written and interesting to read and contains well prepared figures summarizing the main physiological mechanisms described in the review.
I have only the some minor remarks.
It would be good to add to the manuscript a table summarizing methods for microbiota modulation. It could be done in a similar way like it is done in papers describing cognitive behavioral techniques for treatment of insomnia (please see as an example Baglioni et al. The European Academy for Cognitive Behavioral Therapy for Insomnia: An initiative of the European Insomnia Network to promote implementation and dissemination of treatment. J Sleep Res. 2020;29(2):e12967).
Following two sentences are contraindicatory “Using the forced swimming test, an animal model of depression, it was concluded that bilateral destruction of the SCN had a protective effect during the induction of behavioral despair [70, 71]. Similarly, rats exposed to constant light exhibit increased depressive-like behavior accompanied by a loss of diurnal rhythms in physical activity and melatonin and corticosterone levels [69].”
The first sentence indicates that destruction of SCN resulting in deterioration of circadian rhythm protects against depressive symptoms in animal model, the second sentence indicates that consent light resulting in deterioration of circadian rhythm increases depressive symptoms.
Author Response
Authors’ Response to Reviewers Comments
Ref.: IJMS-2334156
Title: Circadian disruption and mental health: the chronotherapeutic potential of microbiome-based and dietary strategies
Prof. Dr. Vasso Apostolopoulos
Special Issue "Chronic Diseases: Gut-Brain-Immune-Microbiome Axis"
International Journal of Molecular Sciences
Dear editor,
Thank you for your kind response on April 7 regarding our manuscript allowing us the opportunity to submit a revised version. On behalf of all the Authors we are very grateful for the kind comments provided by the editor and each of the external reviewers which have helped us to improve the clarity of the manuscript. According to the suggestions, we have incorporated the changes proposed (highlighted in the text), incorporated the Table proposed by Reviewer 1, the changes proposed by Reviewer 2 and thoroughly revised the manuscript. We hope that the modifications made will help you for considering the publication in International Journal of Molecular Sciences. The final version is enclosed and point-by-point responses to the comments are listed below.
Reviewer Comments:
Reviewer #1:
We thank the reviewer for the valuable comments
1. It would be good to add to the manuscript a table summarizing methods for microbiota modulation.
Response: It was done (Table 1. Page 10, line 436)
2. Following two sentences are contraindicatory “Using the forced swimming test, an animal model of depression, it was concluded that bilateral destruction of the SCN had a protective effect during the induction of behavioral despair [70, 71]. Similarly, rats exposed to constant light exhibit increased depressive-like behavior accompanied by a loss of diurnal rhythms in physical activity and melatonin and corticosterone levels [69].”
Response: We have modified the paragraph as follows:
There is also experimental evidence about the association between disrupted circadian rhythms and depressive-like behaviors [68,69]. “Using the forced swimming test, an animal model of depression, it was concluded that bilateral destruction of the SCN had a protective effect during the induction of behavioral despair [70, 71]. On the other hand, (Page 7, lines 323-324).
However, it is difficult to extrapolate the findings of animal tests to human subjects as the circadian system of diurnal and nocturnal mammals differs. (Page 8, lines 330-332)

Reviewer 2 Report
The authors provide an extensive and comprehensive review on circadian rhythm and mental health also suggesting the potential use of dietary strategy through the modulation of gut microbiota to prevent and treat mental illness. The authors should inprove the conclusions session to further clarify their main message, the final statement appears as not conclusive (or truncated?). The authors can consider to suggest a more specific experimental model directed to a defined pathology (i.e. depression, schizophrenia) to test the proposed hypothesis.
Author Response
Authors’ Response to Reviewers Comments
Ref.: IJMS-2334156
Title: Circadian disruption and mental health: the chronotherapeutic potential of microbiome-based and dietary strategies
Prof. Dr. Vasso Apostolopoulos
Special Issue "Chronic Diseases: Gut-Brain-Immune-Microbiome Axis"
International Journal of Molecular Sciences
Dear editor,
Thank you for your kind response on April 7 regarding our manuscript allowing us the opportunity to submit a revised version. On behalf of all the Authors we are very grateful for the kind comments provided by the editor and each of the external reviewers which have helped us to improve the clarity of the manuscript. According to the suggestions, we have incorporated the changes proposed (highlighted in the text), incorporated the Table proposed by Reviewer 1, the changes proposed by Reviewer 2 and thoroughly revised the manuscript. We hope that the modifications made will help you for considering the publication in International Journal of Molecular Sciences. The final version is enclosed and point-by-point responses to the comments are listed below.
Reviewer Comments:
Reviewer #2:
Thank you for your kind comments about our work. They have helped us to review the manuscript.
1. The authors should improve the conclusions session to further clarify their main message, the final statement appears as not conclusive (or truncated?).
Response: According to your suggestion we have modified the conclusion to improve the clarity.(Page 19, lines 830-835).
2. The authors can consider to suggest a more specific experimental model directed to a defined pathology (i.e. depression, schizophrenia) to test the proposed hypothesis.
Response: We have included a proposed experimental model (Page 15, line 633-640).

Reviewer 3 Report
Summary: In this review the authors Codoner-Franch et al., are trying to establish a connection between gut microbiota and circadian rhythms. The authors are highlighting the importance of research towards chrononutrition to develop effective and safe microbiome and dietary strategies. The authors believe that, elucidation of the altered microbiome chronobiology linked to mental disorders may pave the way for highly innovative novel therapeutic approaches based on microbiome modulation.
Comments:
The authors provided sufficient introduction and discussion regarding the clock genes, central clock, chronodisruption , chrononutrition and nutritional psychiatry etc.,The overall manuscript is very well organised and clearly highlighted the importance of personalized nutrition in terms of reducing the prevalence and burden of mental health disorders.
Author Response
Authors’ Response to Reviewers Comments
Ref.: IJMS-2334156
Title: Circadian disruption and mental health: the chronotherapeutic potential of microbiome-based and dietary strategies
Prof. Dr. Vasso Apostolopoulos
Special Issue "Chronic Diseases: Gut-Brain-Immune-Microbiome Axis"
International Journal of Molecular Sciences
Dear editor,
Thank you for your kind response on April 7 regarding our manuscript allowing us the opportunity to submit a revised version. On behalf of all the Authors we are very grateful for the kind comments provided by the editor and each of the external reviewers which have helped us to improve the clarity of the manuscript. According to the suggestions, we have incorporated the changes proposed (highlighted in the text), incorporated the Table proposed by Reviewer 1, the changes proposed by Reviewer 2 and thoroughly revised the manuscript. We hope that the modifications made will help you for considering the publication in International Journal of Molecular Sciences. The final version is enclosed and point-by-point responses to the comments are listed below.
Reviewer Comments:
Reviewer #3:
The authors are grateful for the constructive comments of the reviewer about the article.